# The Perception of Volleyball Student-Athletes: Evaluation of Well-Being, Sport Workload, Players’ Response, and Academic Demands

**DOI:** 10.3390/healthcare11111538

**Published:** 2023-05-25

**Authors:** Roberto Vavassori, María Perla Moreno, Aurelio Ureña Espa

**Affiliations:** 1Department of Sports Science and Physical Education, University of Granada, Carretera de Alfacar 21, 18011 Granada, Spain; perlamoreno@ugr.es (M.P.M.); aurena@ugr.es (A.U.E.); 2Mixed Institute of Sport and Health of the University of Granada iMUDS, University of Granada, Parque Tecnológico de la Salud, Av. del Conocimiento, s/n, 18007 Granada, Spain

**Keywords:** physical activity, wellness, lifestyle changes, health, monitoring

## Abstract

Physical activity has been shown to improve the health and well-being of students, athletes and the general population, especially when it is properly monitored and responses are evaluated. However, data are mostly gathered without considering a valuable element, participants’ perceptions. Therefore, the objective was to know the perception of volleyball student-athletes when using different monitoring and response tools that assess well-being, workloads, responses to workloads, and academic demands. A qualitative study using semi-structured interviews with female volleyball student-athletes (*n* = 22) was used to know players’ perceptions when using a wellness/well-being questionnaire, session ratings of perceived exertion (sRPE), and countermovement jumps (CMJ), and consider academic demands. Results show that the wellness questionnaire and sRPE increased student-athletes’ awareness of well-being and readiness to perform, improved self-evaluation, self-regulation, and self-demand. However, motivation and overcoming challenges were based on the CMJ. Academic demands affected 82% of student-athletes, altering stress, fatigue, and sleep quality. Nonetheless, sport was seen as an activity that helped with academic commitments. Therefore, the wellness questionnaires and the sRPE facilitated self-awareness and positive dispositions toward self-regulation. Simultaneous intensive academic demands and training can produce mutual positive effects if the variables of physical and mental loads are harmonized in the critical academic and sports periods.

## 1. Introduction

Physical activity and participation in sports have been shown to improve mental health, self-esteem, and stress [1]. Likewise, such improvements have been shown to improve physical health, well-being, and quality of life in the general population and athletes of any age and level [2,3]. In addition, systematic physical activity seems to improve perceptions related to attention, motor reaction, motor coordination as well as emotional stability, memory, and imagination [4]. An improvement in academic demands has also been observed in student-athletes who deal simultaneously with academic demands and participation in sports [5]. However, if physical activity is not properly prescribed and monitored, the benefits can be reversed, affecting health and performance [6].

Therefore, to promote athlete sport adherence, improve well-being and wellness status, and achieve established goals, coaches and team staff should be educated on the monitoring of workloads, evaluating the response of athletes, and interpreting readiness status [6,7]. However, athlete opinions also play an important role in these processes, so multiple initiatives are in place from different areas (technological improvements, progress in planning and development in sports, and health sciences), and through them, it would be interesting to know what athlete perceptions are from the use of different monitoring and response tools. This should be a key point in the training process, trying to adapt it and optimize the needs and demands of the athlete [8].

The ultimate goal of monitoring training loads and athlete responses is to improve performance, but also to detect when prescribed workloads do not reach established key performance indicators, control fatigue during the season, avoid overtraining, and reduce injury risk [7,9,10].

Although there are multiple monitoring and response methods in the literature, a consensus on which measures should be used seems to have not been reached [11,12]. However, a combination of internal loads, external loads, and athletes’ responses is widely recommended due to the psychophysical nature of loads and responses [13,14].

Despite technological advances in the objective and quantitative measurement of workloads [15], asking athletes about their subjective and qualitative perceptions is still of irreplaceable value [8,16]. Some widely used subjective methods were applied to better understand the responses of players to workloads in sports and specifically volleyball, including the wellness or well-being questionnaire [9,17] and the session rating of perceived exertion (sRPE), using the duration of the session in minutes and the CR-10 scale [18] to monitor internal workloads [19,20,21,22]. As a combination of the methods is suggested, for external and objective measures, the countermovement jump (CMJ) is a simple and valid tool with which to determine players’ readiness to practice and compete [23,24] and also because the use of jumps has extensively been used in volleyball [17,25].

In the student population attending high school and university, academic periods are considered high-stress. This stress can be divided into two aspects: a social aspect, from the high-pressure environment of society, and an academic aspect, created by the teaching and academic load. Furthermore, in student-athletes who combine sports and academic demands, physical stress is added to social and academic aspects. However, physical stress is a paramount process the body needs to face to make adaptations and maintain performance in sports [26]. Despite the negative effect sports environments can have on students, motivational climate tasks and participation in sports have been shown to reduce levels of stress and increase academic performance [27]. Still, student-athletes are more vulnerable to increased levels of stress during exams and preseason. 

Although studies show the importance of monitoring training/match loads and controlling players’ readiness to perform, this information is usually solely used by coaches and team members to adapt the training and to prepare the best way to perform in the next match [13]. However, the perceptions of the players are extremely valuable and it is important to ask the opinions of the players in different sports [28,29], and in volleyball [16,30]. Using an approach based on players’ perceptions makes them participants in the process, so they can adapt and regulate themselves if needed, which is closely related to formative evaluation. This type of evaluation promotes self-evaluation environments in which participants are responsible for the content, knowledge, and learning process [31] and in sports, for performance.

To our knowledge, few studies have attempted to gain insight into athletes’ perceptions from monitoring and responses tools. Three studies were conducted in both sexes; one explored the perception of watersport sprinters using a training monitoring system [32], while in another, the use of a wellness app showed an increased perception of self-awareness of wellness in ballet dancers [33], and the third explored the perception of multi-sport student-athletes on the effect of academic demands on performance and lifestyle [34]. However, none of these have been used with volleyball teams, and as has been shown in the scientific literature, ball sports produce different responses than endurance sports and weight-bearing sports do [35]. In addition, female athletes seem to perceive greater distress than male athletes do [35]. However, there have been multiple studies showing how exams and highly demanding academic periods affect performance [36,37], but not many have attempted to know the reasons why athletes feel affected or not by these periods directly from athletes.

Therefore, the objective of this study is to know the perception of female volleyball student-athletes when using different monitoring and responses tools that evaluate well-being, workloads, responses to workloads, and academic demands, helping coaches and players regulate sports and life demands to improve players’ wellness and well-being status, mental and physical health, and generating lifestyle changes to improve on-court and off-court performances.

Specifically, the objectives were to know (1) how evaluations regarding well-being, the training/competition exertion, and the readiness to perform affected the players; (2) the suitability of the tools and/or aspects that might be included in the different methods used in the study; (3) the utility and relevance of the different evaluation tools used in the study; (4) the conflict between academic and sports demands.

## 2. Materials and Methods

### 2.1. Sample

The sampling method used was non-probability and convenience sampling. Composed of student-athletes from a women’s volleyball club (*N* = 22), the group of participants included 11 players from the first team competing at the national level, 9 players from the youth team (U19) competing at the regional level, and 2 players participating in both competitions. The players’ ages ranged from 17 to 23 years old (M = 19.41; SD = 1.74). A total of 18 players were studying for a university degree and/or a master’s degree (some of them also combined their studies with work), while 4 players were studying at the high school level, preparing for university entrance exams. 

Data collection occurred during the entire season (September to May); the first team’s competitions included those in the first league stage and the second league stage. U19 had a non-divided competition format, playing each team twice (one home and one away game). Both teams (first team and U19), typically trained 5 days a week, for approximately 9–12 h on the court and 2–3 h for strength and conditioning with, generally, 1 game per week on Saturday or Sunday (on some weeks, 2 or more games were played in the same week).

All players had previous experience using the scales and the questionnaires, and performing the measures of this study as the same structure was in place the previous seasons. However, for clarification, the authors had a meeting with the players during the pre-season to clearly explain (1) how the wellness questionnaire must be filled in; (2) how to register periods of high academic demands; (3) how to register the duration of the sessions and the RPE; (4) how to use the dedicated spreadsheets, use Excel on their phones and how to refresh information to make accessible to the authors; (5) how to record CMJ for evaluation and how to send videos for analysis correctly through an added tutorial. In addition, a 1-month adaptation period was given to the participants to answer the questions and re-familiarize themselves with each tool and measurement. In a different meeting, only with coaches and team staff, the authors went through the same process, to allow coaches to be able to answer players’ questions if necessary.

Confidentiality and anonymity were guaranteed to all participants. Players were informed of the study and consent forms were signed following Spanish guidelines for scientific research on human beings. As 4 players were minors, specific consent forms were signed by an adult (parents or legal guardians). The Declaration of Helsinki recommendations were carried out during this investigation. For integrity and transparency, this investigation was approved by the research ethics committee of the authors’ university, 2070/CEIH/2021.

### 2.2. Protocol for the Players’ Monitoring of Loads, Responses Measurement, and Academic Demand Data Collection

The intervention protocol consisted of the collection of subjective and internal load monitoring data related to training sessions and matches, internal and external responses of student-athletes to those practices and games, and information about academic demands.

The first author of this study and head of strength and conditioning for the university volleyball teams was in charge of the monitoring and response protocol. Being part of the team staff increases the trust of the coach, as well as the commitment of the players. For the intervention, a dedicated Excel spreadsheet (Microsoft 2013, Redmond, WA, USA) was created to record the information of the players (one tab for the wellness questionnaire that included an additional item for the academic demands information and another tab for the sRPE). Each player had the Excel document downloaded to their mobile phones and connected to a Google Drive (Google LLC, Mountain View, CA, USA) folder, access to which was granted exclusively to the authors of this study. Coaches also shared information with the authors, transferring video recordings of the players’ jump performances via mobile phone for posterior analysis.

### 2.3. Wellness Questionnaire

The responses to training and competition were collected using a wellness questionnaire completed by the players daily and containing 5 items (fatigue, sleep quality, muscle soreness, stress, and mood). Players’ well-being was rated on each item using a 1–5 Likert scale (1 as a worse/lower score), and a total score could be obtained by adding up each item’s score (scores ranging from 5–25 points). A comment box was added to the questionnaire and the players were asked to register periods of high academic demands: exams, project deadlines, exams periods, and dates of increased studying hours. 

### 2.4. sRPE

The sRPE was used to monitor players’ internal training load. After each training session or competition, the players included the duration of the sessions in minutes, the type of session (practice, game, and gym/strength and conditioning), and the RPE following a CR-10 scale provided in the Excel document. Once the information had been filled in, the workload was calculated automatically in arbitrary units by multiplying session duration and intensity. From the data obtained, daily workload, weekly workload, acute-to-chronic workload ratio, and weekly differences were also calculated.

### 2.5. CMJ

The countermovement jump gave objective information about the players’ readiness to practice and compete. Twice a week, 72 h after matches, 48 h before the following competition [25] and after the team on-court warm-up jump measurements were performed, coaches recorded videos of the players performing two CMJs with their hands on the hips [38]. Videos were filmed using an iPhone 5s (Apple, Cupertino, CA, USA) set in slow motion (120 fps). Players’ bodies had to be recorded fully with the camera placed on the ground, so whether or not feet were in contact with the floor could be observed. After each recording session, the videos were sent to the authors of the study, and jump height, power output, and readiness to perform were measured using the MyJump 2 app, a valid a reliable tool [39]. To promote motivation and full effort during jumps, hence maintaining high reliability and validity [40], personal best jumps were notified to the team during the season and a players’ rankings for jump height and power output were shared monthly. 

For a better understanding of the protocol and the information collected weekly, see Table 1.

### 2.6. Data Collection and Instrument

A semi-structured interview technique was used. All the 22 student-athletes participating in this study took part in the interview process at the end of the season, when the monitoring and response data collection was completed. The main reason for using this technique was to better understand the perceptions of the players when assessed on their well-being and perceived exertion; improve the tools used during the season by knowing players’ suggestions; obtain players’ tools preferences; understand how academic demands conflict with sports demands. The interview was structured in 5 blocks: wellness questionnaire questions, sRPE questions, CMJ questions, tools preference questions, and academic demands questions (Table 2).

The second author of this study, with extensive experience in semi-structured interviews and qualitative analysis, performed and recorded the interviews, and acted as an active listener during the questions, in accordance with established recommendations [41]. The interviews were recorded in an isolated meeting room in the training venue, using the recording app for a 5th generation iPad (Apple, USA). The duration of the interviews ranged from 9.31 to 29.20 min (M = 16.19 min). A verbatim transcription of the content of each interview was performed for an accurate and complete record of the data collected.

### 2.7. Data Analysis

Once the data entry process was completed, interviews were transcribed verbatim and a thematic analysis was implemented [42,43]. From the interviews’ data, a reflexive thematic analysis (RTA) was also put in place for a deeper engagement with the information obtained from the players [44]. To follow current qualitative research guidelines, a six-step approach was used in the data analysis [43,45]. In the first step, the transcripts of the interviews were read several times until a complete understanding of what the players were trying to express was reached. Step two focused on the labeling of the relevant raw data; labels were then coded and the initial themes were generated. Coding was reviewed by the authors of this study and discussed until they reached a consensus. Similar codes were merged and refined to better understand what the players were trying to say and simplify the selection of themes. The third step consisted of finding similar ideas in the interviews, so patterns could be created. Step four assessed the different themes to make them clearer and more concise. If needed, some themes were combined into new themes or sub-themes, and in the situation of a lack of sufficient data or an inability to integrate them into other themes, the authors decided to remove those themes. The fifth step focused on creating titles for each theme and a clear description of what they involved. The relationship of each theme and sub-theme with the research questions was also explored in this step. The goal of the last step was to create a link between what the data was trying to tell us and the perception of the players.

### 2.8. Trustworthiness

The use of interviews for data collection is a key element in qualitative studies in sports and requires specific competencies that researchers need to be skilled in and prepared to have. Experience in interview techniques and qualitative research ensures the credibility and integrity of the data collection process [46]. In our study, the interviewer had several years of experience in the use of semi-structured interviews, implementing qualitative analysis, in volleyball teaching and volleyball coaching. Therefore, it can be stated that the level of credibility in the use of the data collection technique was high.

Furthermore, the authors of this research hold the highest volleyball coaching qualifications (level 3), two of them having the highest academic volleyball qualification (Ph.D.) and sports qualification (international coach level 2), and over 20 years of experience in teaching and/or training volleyball. Hence, during the discussion of categories and subcategories among the authors, we were confident to confirm that the level of expertise in the topic is reasonably high.

Lastly, transcripts and coding interpretation followed a “critical friends process”, where the lead author discussed the data interpretations with the other researchers to ensure high-quality research by adding value and rigor, despite knowing that other possible data interpretations might have existed [47].

## 3. Results

The structure of the results follows the order of the interview questions order, divided into three blocks for each tool used by the players, one for academic demands and one for tool preference. Each block will be supported by a frequency table and excerpts from the interviews to help the reader understand better the results.

### 3.1. Wellness Questionnaire

The results generated subcategories of a causal attribution between the tool and the players. A perception of awareness was observed in 100% of the players (Table 3). 

Specifically, 50% of the players experienced neutral awareness and the remaining 50% had positive awareness. In neutral awareness, the players perceived that filling in the questionnaire made them think about how they felt, and this can be seen in the following excerpt.


*It made me realize, it made me ask myself, how am I? What is the workload that I have? Or if I have slept less, try not to make it affect the training. Or, If I had a personal issue, it might have influenced my status as well. But in general, it made me aware of what I am feeling.*
(Player 3)

On the other hand, positive awareness shows a deeper effect on the players, perception of the tool being really helpful. This includes neutral perception, but additionally includes a positive response, showing to the interviewer that it clearly helped them.


*I like it! because I was thinking about how I really felt. Because, otherwise, you do not know until you do not see it. How tired are you? How much have I slept? I suppose if you do not do the questionnaire, you do not think about it. These are the reasons why I liked the wellness questionnaire.*
(Player 21)

Additionally, also helped to better understand themselves better and the importance of self-evaluation.


*It has been a way of knowing me better during the season, and saying: oh! That is why I was more susceptible… so then, I would write it down in the questionnaire, and as we do so many things during the day, until we do not stop and think, and self-evaluate we do not know the real motive that we feel in a certain way.*
(Player 6)

There were some factors that players pointed out that were particularly affecting them. One of them was fatigue, expressed by 59% of players.


*I felt very fatigued at the end of the training and I felt… not being the same. Seeing it written in the questionnaire shows how closely related it is [the questionnaire] to training and fatigue.*
(Player 1)

The next factor was academic demands in 50% of the players, personal issues, and sleep (41% and 32%, respectively). Mood state was related as a factor affecting wellness status in 23% of interviewees and 14% and 5% of players also mentioned stress and muscular soreness, respectively.


*There were also other factors, such as when we had exams and other academic demands.*
(Player 9)


*At the end of the day, I was able to see my evolution and say: When we exercise more, I am more fatigued. Or I do not sleep that much or the other way around. Then, it is true that it helps you to follow up on how you feel. And what is better or worse for me.*
(Player 10)

One of the interview questions asked for suggestions to make the tool more complete and solve some personal or specific interests. A total of 23% of players suggested the inclusion of a question regarding nutrition into the questionnaire. Other suggestions were related to injuries or physiotherapy, making up 14%, and their period, work, and other activities, making up 5% of the players for each topic.

### 3.2. sRPE

A causal attribution between neutral and positive awareness was found by completing the sRPE. In total, 21 players experienced awareness of some type, and specifically, 64% were neutral and 32% positive (Table 4).

Similarly to the wellness questionnaire, a positive response was established using the sRPE.


*By filling in Excel, you realize how it affects you, otherwise, you would not know, you do not think about that [RPE and intensity] when you go to practice, at least is what I felt years ago when we were not using this tool, it was not even crossing my mind.*
(Player 1)

In the next excerpt, an example of positive awareness can be observed, as the player finds it important and also states her rationale for this perception.


*To me, this is very important because many times we feel fatigued and we do not know why. Perhaps, being able to write down how we feel and see each week’s intensity numbers, we can adapt and train harder or be more relaxed the following day or week; it can help us a lot to improve.*
(Player 14)

The sRPE adds another subcategory not observed in the wellness questionnaire. Of the 22 players, 16 (73%) perceived self-regulation, self-demand, and self-evaluation from recording their answers. Self-regulation is explained as adapting their intensities based on previous training sessions or training weeks, and we can see this in the excerpt from Player 14 above. Players would regulate their efforts by increasing intensity if they gave a low number and reducing intensity if they felt they rated previous sessions too high. Self-demand can be seen in some players as giving their best, and not agreeing with minimal effort.


*I saw that during training I was not well. I saw many low numbers. Then, I have to do something, I need to change, right? Because if for me the intensity was not high, maybe it was because I was not doing it correctly.*
(Player 3)

Self-evaluation is related to the management of critics of their performances and is used to adapt and manage certain life and sports situations. It could be related to self-criticism; however, self-evaluation has positive connotations to help players improve and promote performance perceptions and well-being.


*And you finish practice with a bad feeling and if you do not say it or express it to anyone else… and you felt that you did not do anything, you have come and lost your time… writing it down helped me let the steam off and say: I am going to put myself a 2 (out of 10) because I felt I have not done enough.*
(Player 2)

The main factors that affected sRPE reported by the players were fatigue (59%), mood state (18%), and academic demands (14%). Fatigue was seen as players feeling more tired than normal after certain training sessions and competitions, and they were able to detect that feeling of tiredness via observing a higher number on the RPE CR-10 scale and in days after intensely rated practices. Using the sRPE tool, we can also observe how players were able to better understand the possible reasons why they felt more fatigued and self-regulate in the future.


*If on a certain day, you felt more tired, evidently for me the training was a 7, and another day that I was fully rested…maybe a would have been a 4.*
(Player 21)

The mood state also affected sRPE; a worse mood state caused the players to rate their sessions higher, and vice versa. Mood state seems to be affected by external factors not related to the team, and academic demands appear to be very involved in mood changes. Similarly to fatigue, completing the sRPE tool created awareness in the players.


*At the end of the training used to vary, it was different depending on your mood. It was like some days… you are happier and you practice better with more energy. And maybe on a day that you are sad or you are not having a good day, you have exams or other things… you are not as intense. But by measuring with Excel, you can see it better.*
(Player 8)

Academic demands seem to affect training intensity and hence, workload. During periods of exams, days spent studying or with assignments for university and high school, the players rate sessions as harder than normal. Other factors affecting sRPE were stress, mental fatigue, and other activities, each of them observed in 9% of the players. The following statement shows the connection between fatigue, mood, academic demands, and personal problems.


*I was able to relate that the following day I was more tired when going to class because of the training intensity. Or, for example, when I was moodier, especially because I had exams. Or I had a bad day. Normally, I used to rate those training days as more intense. And of course, it is because you get more tired, and you are more… blocked that day. You say: I just want to go home, I do not want to continue. Thus, the intensity increased. Therefore, in those situations, I realized… that they were connected.*
(Player 15)

Suggestions were also requested for the sRPE, and 59% of the players thought nothing else should be added and expressed that they would keep the tool as it is. Some players indicated that they would differentiate between physical and mental loads in training (23%). These players felt that the mental fatigue generated by sport, academics, and other life factors were affecting their well-being and performance, and suggested the deeper monitoring of mental factors. On the other hand, 14% revealed that the scale was complex, especially differentiating from one number to another, and also an adaptation period was needed to use the tool properly.


*Perhaps a single number was limited. Because sometimes at the beginning of the session, I was feeling very well but then my intensity was 4 because the intensity was low, but I was feeling great. So, I don’t really know how I would do it, but maybe using one number only is too little. Sometimes I struggled to put a number from 1 to 10 for the intensity of the session.*
(Player 17)

Other suggestions such as taking into account contextual factors (matches, home or away, and distances traveled) (5%) and comparing players’ RPE vs. coaches’ RPE (5%) were given during the interview process.

### 3.3. CMJ

The CMJ measurement was perceived as positive, increasing motivation, competition, and overcoming challenges by 86% of the players (Table 5).


*Measuring jump heights made us try to beat other players on the team, as the coaches told us who was first, and second and how we were evolving. It was a nice challenge to face and it made us set goals to overcome. It was great and very relevant.*
(Player 14)

Some players experienced an evolution and improvement in their scores, making the intervention period more positive (18%).


*You try hard, because when you start seeing improvement… for example, I did not start very well, I was not standing out, I was in the bottom half of the rankings. And suddenly I started to improve and go up and up.*
(Player 2)

However, the same amount of players (18%) had the opposite perception, experiencing a negative evolution/improvement. In addition, a negative response to the tool perceiving it as not helpful was observed in two players (9%). 


*In the middle of the season, I stopped seeing my name in the personal bests. It is true that at the beginning we were very motivated. But then I reached my peak and maybe I was more fatigued, studying, etc. At the end of the season, I felt the jump like… I will do it and that is it. But that motivation faded because I had to make a perfect jump to beat myself…*
(Player 6)

Similarly to what they did with other questions in the interview, players revealed which factors were affecting the CMJ; some were positive, and others negative. One of the negative factors was fatigue in 27% of the participants; players perceived a decrease in jump height and power output when tired.


*Especially the feeling of being recovered, because when you drag fatigue from the whole week, or, for example, on Fridays, I am sure that we were jumping much less than on Mondays or when measuring at the beginning of the week.*
(Player 7)

A total of 14% of players experienced that the jumping technique was different than the one used in volleyball. During measurements, players would start standing (with no approach) and with no arm swing. Although this is a common and validated protocol for CMJ measurement, players thought they could jump more with another technique.


*I think I jump more with approaching steps. Well, everyone does, but… I think this way would have motivated me more.*
(Player 9)

Academic demands affected 9% of the players. Accumulation of fatigue, stress, lack of sleep from exams, and high-volume academic periods seem to have affected jump capacity. 


*I improved a couple of times in a month, but then I got stuck. I do not know exactly why. I may have been fatigued, and especially after Christmas and the exams, made it feel even more.*
(Player 22)

Contrastingly, among the effects making a positive impact on CMJ, 27% revealed the similarity of the task with the sport, especially perceiving jumping as a key performance indicator in volleyball and thinking the more they jump, the better the on-court results.


*I liked it because it was a challenge, in volleyball jumping is very important and obviously, the more you jump the more you can clear the net and increase your performance, right?*
(Player 14)

Some players (18%) established a positive relationship between the work performed in the gym and during strength and conditioning sessions with the CMJ. The perception was that increasing strength and power in fitness sessions would translate into increased jump height. Additionally, 5% of interviewees pointed out the relationship between the CMJ and the wellness questionnaire, interpreting that some items of the questionnaire made them jump more.

A suggestion section was included for this tool as well. A total of 59% would have been interested in measuring speed, acceleration, and agility. The perception was that measuring these physical characteristics can improve readiness to practice and compete. Different variations/modalities of jumps were recommended by the 23%, specifically about measuring jump with approach and arm swing as stated by Player 9 previously.


*Maybe speed or reaction. I think it could be interesting to evaluate. Especially speed. Or try to be more agile. Like when we need to block. All these might be good for volleyball and may be more interesting than jumping alone.*
(Player 7)

Measuring strength was suggested by 18% of players and endurance assessments by 14%, both of these suggestions being pointed out as a way to enhance their levels on the court. Less observed suggestions were coordination, nutrition, and ball perception (decision-making regarding the ball and its trajectory) with 5% suggesting these each.

### 3.4. Academic Demands

Academic demands affected players’ well-being and readiness to practice and compete in 82% of cases (Table 6).

The most reported reason affecting players was stress, seen in 41% of players. Stress is generated during periods with exams and preparation for those tests and appears to modify the status of players on and off the court.


*In the end, during exams, we are very stressed and burdened, which affects our performance. Personally, it affects me and it is noticeable.*
(Player 10)

Fatigue was mentioned by 41% of players; during highly demanding academic periods players felt more tired than usual, worn out, and extremely fatigued.


*It affects a lot. Because in the end, you are tired from studying all day… In general, I think it happens to everybody, because you see it in the person next to you, that she is fatigued too.*
(Player 9)

Sleep quality was mentioned by 23%; the lack of sleep being mentioned due to the increase in studying hours made the players feel restless, therefore affecting well-being, sport, and academic demands. 


*Well, during exams the sleep quality was way worse, I was sleeping less. The intensity in training was much lower because we were fatigued, and I noticed that a lot.*
(Player 12)

Deconcentrating and mental factors affected 18% of cases each. The players defined deconcentrating as not being fully immersed in practice sessions as their thoughts were on future academic demands.


*In training, I was thinking more about the exam I was going to have the following day than in the training itself, and that was very noticeable in how I was performing, it was not a good training session at all.*
(Player 1)

The mental factors were explained by the players as different than the physical aspects, opposite to body or muscular fatigue. The extract below exemplifies what the mental factors meant to the players.


*I am not that agile mentally. I am tired, for example, sometimes I was getting home and thinking: Physically I am ok but mentally I am KO.*
(Player 21)

There are other less common reasons players are affected, these including being self-demanding (14%), physical factors 9%, family pressure 9%, and lastly, perceiving the academic year as difficult 5%.

A total of 18% of players perceived that academic demands did not affect them. As for the reasons not affecting the players, these were primarily self-organization (14%) and the ability to schedule academic and sports commitments at the same time to avoid one affecting the other.


*It affects me very little or not at all. I have always been able to combine the two. To be honest I can’t study and miss practice. I need to practice and then study.*
(Player 3)

A total of 9% of players had the perception that the academic year was easy. A commitment to the team and the desire not to let other team members down by missing practice to study was found in 5% of players, which is the same as the amount of players with a low number of exams.


*I am studying teaching and it does not really affect me much. The degree is pretty accessible. If I put in the hours I need, it will not affect volleyball.*
(Player 11)

From the interview process and during the questions about how academic demands affected their performance, an inverse topic regarding the potential effects of being a student-athlete on academic performance came up without being directly asked to the players and which we think is of extreme interest. A total of 45% of the players saw volleyball as an escape route, a way of freeing their minds and helping them with their studies. Sometimes, there is also a need to evade academic demands as a form of academic enhancement. At the same time, participation in sports improved time management (36%).


*In other seasons I was at home thinking: I could be training and I would have cleared my mind and that would have helped me memorize. So, for me… to improve my academic performance, I need to come here (practice) first. It is my escape route.*
(Player 3)

The last two factors are directed at coaches, with the suggestion to take into account training and competition volume, to help the players dedicate themselves to volleyball and studies, increase their well-being, and also perform at their best on the court and in the classroom (14%) as well as to keep in mind players position (5%) for the same reasons.

I *also have to say, maybe, being the setter, during practice I have to think. Think about which is the best play… like I have to use my brain a lot. And if I spend 6 h studying, plus lessons and projects, I feel it.*(Player 21)

### 3.5. Tool Preference

One question during the interview was about selecting the tool that helped the players better understand their readiness to practice and compete and also their well-being. Some players answered multiple options as they felt a combination of tools made their perception better than a single one did. The result showed that 73% of the players selected the wellness questionnaire, followed by sRPE with 45% of players and the CMJ (41%). Two players (9%) perceived that using all three tools together helped them the most. Nine players selected the combination of two tools (41%). Finally, 11 players (50%) exclusively picked one tool.

## 4. Discussion

The authors opted for an open attitude, without prefiguration or expectation in the approach of this study to avoid influencing the reach and value of the players’ own opinions [8]. Although there is an extensive background in load monitoring and management, we did not find records on the perception of self-evaluation in student-athletes. Thus, the methodology used was qualitative and the research strategy was inductive. Hence, the results subsequently orient us towards seeking more friendly theoretical frameworks.

A response that appears consistently in the results of this study is how the self-evaluation of the well-being of the players increased self-awareness, establishing the foundations for the education of self-regulation and the better integration of training into their lifestyle. On the other hand, the sRPE allowed them, also, to establish judgments, including, firstly, judgments about the value of training (directed to others), and secondly, judgments about the value of their own performance (directed to themselves). Therefore, subjective awareness of physical status determined the quality of the training object, as well as the object of personal performance, in terms of their beliefs and motivation frameworks. It is mainly displays of self-awareness and its implications that draw our attention regarding potential theoretical frameworks of reference.

Usually, athletes with an established culture of training and performance have clear references with which to evaluate the quality of a training session, the team’s achievements, and their own from an objective point of view, thus seeing themselves as the object of their judgment. In contrast, from a phenomenological perspective, the internal perception of the body directly communicates the judgment of perceived objects (in our case training and performance).

During the development of the philosophical problem of phenomenology, different disciplines found in its foundation a form of practical action in clinical intervention, such as psychology, psychiatry, or nursing [48,49,50]. In this practical domain, phenomenology is used as a method to better serve patients, and it should disregard the more purist and controversial elements of philosophy [50]. Therefore, this interest in applications could be extended to sports training.

The results of this study show that players’ self-awareness in assessing their well-being and perceived exertion put them in a position to develop self-regulatory mechanisms. Self-regulation has been widely reported as a quality that positively affects athletic preparation and performance, but has also been shown to be important for successful recovery [32,33,51]. It was also reported that the ability to self-regulate and, specifically, to regulate emotions reduced stress, increased mental strength, and produced a better perception of mental health [52]. Furthermore, self-regulation helped in being proactive in pain and sleep control [33] and possibly, as shown in our results, in fatigue, mood state, and muscle soreness. The ability to identify one’s own current state, through the self-monitoring of thoughts, feelings, and behaviors, to successfully regulate them is highlighted [51].

From the ecological meta-theory, self-regulation is part of the questioning of the social cognitive theory. According to Carvalho and Araújo [53], the social cognitive perspective sees cognitive processes and intentions as the cause of behavior, in a linear and deterministic way, separating the environment and the person, while the ecological dynamics perspective understands them as emergent temporal dynamic processes, which limit the way the performer perceives and acts in the environment. Zimmerman [54] revised Bandura’s model [55] to apply it to the scope of academic learning. Zimmerman also assigns the essence of the triadic model of social cognitive behavior to the result of self-generated influences as well as externally generated ones, but in asymmetric reciprocity, and not unidirectional, nor stable over time. In our opinion, it is difficult to interpret linearity and determinism in this statement. Perhaps, the main difference between social cognitive theories and ecological theories is the existence or not of representation.

Studies on self-regulation identified two distinct recovery needs: detachment from sports-related topics and mental rest. The first can be obtained through involvement in mentally demanding activities (studying and talking with friends, among others). The second is through involvement in undemanding mental activities (listening to music) [51].

Regarding detachment, the players interviewed in our study reported that sport was an escape or form of evasion from their academic demands. Sport helps them with their academic demands by relaxing them, clearing their minds, and focusing on other things, so that they can resume their academic commitments in a fresher state. From our point of view, this description fits perfectly with the concept of disengagement of high-performance athletes from sports [51]. The dual perspective, on the one hand, of the student participating in intense sporting activities and, on the other hand, of the athlete studying intensively, finds a parallelism in terms of the mutual benefit of both activities [27].

Additionally, about mental workload, our results draw our attention to some players that mentioned being more affected by stress and having a poorer quality of sleep during periods of high academic demand. This is consistent with the results of a systematic review [56]. Similarly, other studies have shown a conflict between sports and academic demands [36,37]. Furthermore, some players in our study highlighted the need to include specific questions on mental fatigue in the questionnaires. Physical, cognitive, and emotional components interact in the outcome of fatigue, and questionnaires assume their role as black boxes due to the difficulty of establishing quantifiable causal relationships. However, assessing mental fatigue and creating lower-volume, physical intensity-based training sessions with less cognitive and emotional demands could be a viable alternative with which to finally bring both sides together.

Although self-evaluation has made it possible to assess one’s own performance, the extent of the players’ responses does not support the determination of kindness in these self-judgments. The theoretical model of self-compassion consists of three antitheses: self-kindness versus self-judgment, common humanity versus isolation, and mindfulness versus over-identification [57]. In self-compassion theory, attitudes of self-criticism, self-indulgence, self-esteem, and perfectionism are mutually exclusive.

Self-awareness could be improved by education aimed at self-compassion. By rethinking self-criticism, it reaches its potential in sports [58,59]. The impact of self-compassion on well-being is evident, as it improves nutrition, sleep, attention, mood state, confidence [59], interaction with others, responsibility, resilience, resistance, and acceptance of failure as an opportunity.

In our study exists a third tool (CMJ) with which to observe the objective response of players to training and competition loads. This tool provided a small competitive component using a CMJ ranking system that affected the players’ motivation and thoughts. Some players focused on the process of improving their jumping ability by mastering the task, while other players focused mainly on ranking and performing better than other teammates, as has been observed in other studies [60]. Self-motivation and personal growth theories show that people can be intrinsically motivated (task/mastery oriented) or extrinsically motivated (ego/performance-oriented) [61]. Elliot and Church [62] extended the theory and introduced a proposal with three categories based on mastery and two levels of ego motivation: performance approach and performance-avoidance. Performance approach (positive competence judgment) along with mastery also plays a key role in self-regulation and motivation. The reflection of our players points to the predominance of an ego-oriented performance approach.

Of course, this study has some limitations that should be taken into consideration when extrapolating it to other settings, students, and teams. As the entire sample belongs to the same club and academic environment, there may be culture-specific traits that could vary in other environments, in addition to the differences that may occur according to sex and age. Especially, caution should be taken when transferring our results to male athletes, and other ages, as they could be affected by both variables. This study’s sample is one in which the main objective is the academic education of the participants, although they train habitually and intensively, in the case of people who prioritize sports performance; although they are studying, it seems plausible to expect some differences in their responses. Therefore, extension to other populations could be an interesting line of research.

Other research objectives derived from the results of the present work could be, on the one hand, to analyze the effect of proposals that harmonize mental workload with periods of high academic demand, based on the inclusion of mental workload control tools. On the other hand, these could be to investigate the effect of interventions aimed at education on self-compassion.

The most relevant implications for practice that our results can bring to this field of study are the use of phenomenology as a method that could bring value to research on self-regulation in recovery from sports efforts. The results also encourage experimental research on the best possible interaction between academic performance and sports performance.

## 5. Conclusions

The tools used in this study are widely consolidated and validated as load and training response assessment instruments. The novel qualitative approach that we have used has allowed us to discover the added value of these tools. The categories most frequently expressed in the athletes’ responses during the interviews were awareness, self-regulation, and motivation. The overall interpretation of the responses places the self-awareness of the state of athletes and their performance in a position to create the necessary conditions for the effective development of self-regulation. 

The intensive parallel dedication of academic studies and sports training, although it may create some areas of conflict, presents complementary qualities of high value if coaches and athletes know how to manage the variables of training adequately.

## Figures and Tables

**Table 1 healthcare-11-01538-t001:** Example of a week with a game on Saturday.

Day of the Week	Session Type	Measurement/Method	Timing
Monday	Training	Wellness questionnaire Academic demands	Before Practice Before Practice
	Gym + court	sRPE	After Practice
Tuesday	Training	Wellness questionnaire Academic demands	Before Practice Before Practice
	Injury Reduction + court	CMJ sRPE	After Warm up After Practice
Wednesday	Training	Wellness questionnaire Academic demands	Before Practice Before Practice
	Gym + court	sRPE	After Practice
Thursday	Training	Wellness questionnaire Academic demands	Before Practice Before Practice
	Injury Reduction + court	CMJ sRPE	After Warm up After Practice
Friday	Training Injury Reduction + court	Wellness questionnaire Academic demands sRPE	Before Practice Before Practice After Practice
Saturday	Game	Wellness questionnaire Academic demands sRPE	Before Game Before Game After Game
Sunday	Day Off	Wellness questionnaire Academic demands	In the morning In the morning

**Table 2 healthcare-11-01538-t002:** Semi-structured technique interview questions.

Wellness Questionnaire Questions	
Q1	Self-assess your wellness status (questions before the training), what did it mean to you?
Q2	Do you think that there are other questions that affect your wellness status during the season that were not included in the questionnaire? If yes, indicate what these are and the relevance given to each of them.
**sRPE questions**	
Q3	Self-assess the intensity of the session via RPE and register the session duration (questions after the training), what did it mean to you?
Q4	Do you think that there are other questions that affect your training and match intensity during the season that were not included in the questionnaire? If yes, indicate what these are and the relevance given to each of them.
**CMJ questions**	
Q5	What is your opinion about the CMJ measurement and how do you face it?
Q6	What other abilities/activities/tasks do you think would be useful to know your readiness status to train/compete?
**More useful tool question**	
Q7	What information we have asked during this interview (wellness questionnaire, sRPE, CMJ), do you consider to be more useful to know your status to train and compete?
**Academic demands question**	
Q8	How do academic demands affect your sports performance?
**Other questions**	
Q9	Anything else you would like to add?

**Table 3 healthcare-11-01538-t003:** Descriptive analysis of the perception of completing a wellness questionnaire.

Categories	Subcategories	Frequency	Percentage	Players
Wellness questionnaire causal attribution	Neutral awareness	11	50%	1/3/4/6/7/9/15/17/19/20/22
	Positive awareness	11	50%	2/5/8/10/11/12/13/14/16/18/21
Elements or factors affecting well-being	Fatigue	13	59%	1/2/7/8/9/10/13/14/15/16/19/21/22
	Academic demands	11	50%	1/2/5/7/9/10/14/16/18/19/21
	Personal issues	9	41%	1/3/4/5/9/13/14/18/22
	Sleep	7	32%	1/2/3/10/15/21/22
	Mood state	5	23%	2/8/17/18/22
	Stress	3	14%	10/14/22
	Muscular soreness	1	5%	9
Suggestions	Nutrition	5	23%	1/7/11/15/16
	Injuries/Physiotherapy	3	14%	4/16/17
	Period	1	5%	6
	Work	1	5%	19
	Other activities	1	5%	19

**Table 4 healthcare-11-01538-t004:** Descriptive analysis of the perception of completing the sRPE.

Categories	Subcategories	Frequency	Percentage	Players
sRPE causal	Self-regulation/self-demand/self-evaluation	16	73%	1/2/3/4/5/7/9/11/13/14/16/17/18/19/20/22
attribution	Neutral	14	64%	1/2/3/4/5/7/9/11/12/13/15/18/20/22
	Positive	7	32%	8/10/14/16/17/19/21
Elements or	Fatigue	13	59%	4/6/7/9/10/13/14/15/18/19/20/21/22
factors	Mood state	4	18%	7/8/15/20
affecting	Academic demands	3	14%	3/8/15
sRPE	Stress	2	9%	1/13
	Mental fatigue	2	9%	6/22
	Other activities	2	9%	13/19
Suggestions	Differentiate physical and mental training	5	23%	4/6/9/11/16
	Adaptation/complexity	3	14%	6/12/17
	Players RPE vs. coach RPE	1	5%	18
	Contextual factors	1	5%	21

**Table 5 healthcare-11-01538-t005:** Descriptive analysis of perception of completing the CMJ task.

Categories	Subcategories	Frequency	Percentage	Players
CMJ causal attribution	Motivation/competition/overcoming challenges	19	86%	1/2/3/4/5/6/9/10/12/13/14/15/16/17/18/19/20/21/22
	Feeling of a positive evolution/improvement	4	18%	2/3/10/21
	Feeling of a negative evolution/improvement	4	18%	6/9/11/18
	Negative response to the tool	2	9%	8/11
Elements or factors affecting assessment performance	Fatigue	6	27%	2/6/7/19/21/22
	Task similar to the sport	6	27%	1/2/13/14/21/22
	Relationship between strength and conditioning and performance in the task	4	18%	1/2/9/20
	Task different to the sport	3	14%	9/11/19
	Academic demands	2	9%	6/22
	Relationship between the wellness questionnaire and CMJ	1	5%	7
Suggestions	Speed/acceleration/reaction	13	59%	1/2/4/6/10/11/12/13/15/16/18/20/21
	Other jump modalities	5	23%	8/9/12/18/19
	Strength	4	18%	5/7/11/19
	Endurance	3	14%	1/16/22
	Coordination	1	5%	5
	Ball perception	1	5%	10
	Nutrition	1	5%	17

**Table 6 healthcare-11-01538-t006:** Descriptive analysis of the effects of academic demands on performance and sport performance on academic demands.

Categories	Subcategories	Frequency	Percentage	Players
Effect of academic demands on sports performance	Affects	18	82%	1/2/4/5/7/8/9/10/12/13/14/15/16/17/18/19/20/21
Reasons for effect	Fatigue	9	41%	2/4/7/9/12/13/16/20/22
	Stress	9	41%	2/5/8/10/14/15/17/18/22
	Sleep	5	23%	4/7/12/18/20
	Mental	4	18%	2/13/19/21
	Deconcentrating	4	18%	1/5/7/15
	Self-demand	3	14%	2/5/14
	Physical	2	9%	7/13
	Family pressure	2	9%	1/14
	Perception of academic year as difficult	1	5%	1
Effect of academic demands on performance	Does not affect	4	18%	3/6/11/22
Reasons for lack of effect	Self-organization	3	14%	3/6/22
	Perception of academic year as easy	2	9%	11/22
	Low number of exams	1	5%	6
	Commitment	1	5%	3
Effect of sports performance	Sport as evasion	10	45%	1/2/3/6/9/16/14/18/19/20
on academic demands	Time management	8	36%	1/3/5/6/10/11/14/20
	Consider training volume	3	14%	5/12/15
	Consider players position	1	5%	21

## Data Availability

The data presented in this study are available upon request from the corresponding author.

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
