# Peer review of "The Perception of Volleyball Student-Athletes: Evaluation of Well-Being, Sport Workload, Players’ Response, and Academic Demands"

_healthcare, 2023, doi:10.3390/healthcare11111538_

Round 1
Reviewer 1 Report
The objective of the paper is to study the perception of volleyball student-athletes when using different monitoring and response tools that assess well-being, workloads, responses to workloads, and academic demands.
The abstract is well-written and the purpose of the research and some conclusions are clearly formulated. The topic is a subject of intensive research in scientific literature.
Keywords reflect the content of the paper but the first letter should not be capitalized according to the journal’s template.
In the Introduction, the most important notions used in the paper such as training loads, athlete responses, load monitoring etc. are mentioned briefly. This section plays the role of a literature review as well. The section is well-written and the problem and the objectives are explained clearly. I have a few following critical remarks to the Introduction. First, in the first sentence a reference about the notion of ‘stress’ is given and at the end of the sentence two more references are presented ([2],[3]). It is not clear why would a separate reference about the ‘stress’ should be given and not references about the other notions. I recommend to the authors to revise this sentence either by removing source [1], or by adding more source for each notion.
Second, it is not clear why the authors have selected volleyball student-athletes and why the athletes are all females? The authors should state clearly why the chosen sport is volleyball and then refer to similar studies in other sports as a comparison to the present paper and they should address the question of possible gender skewness of the results. For instance, in the paper:
Nyhus Hagum C, Tønnessen E, A I Shalfawi S. Progression in training volume and perceived psychological and physiological training distress in Norwegian student athletes: A cross-sectional study. PLoS One. 2022 Feb 4;17(2):e0263575. doi: 10.1371/journal.pone.0263575. Erratum in: PLoS One. 2022 Mar 28;17(3):e0266313. PMID: 35120187; PMCID: PMC8815906.
a cross-sectional study examines self-reported weekly training volume and perceived training distress in Norwegian student athletes according to gender, type of sport, school program, and school year. The authors recommend that practitioners use a conceptual framework to periodize training and monitor training distress in student athletes, particularly in females, to preserve physiological and psychological well-being and ensure a progressive training overload leading to positive performance development.
Another important paper, that studies the problem is:
Gomez, J.; Bradley, J.; Conway, P. The challenges of a high-performance student athletes. Ir. Educ. Stud. 2018, 37, 329–349.
In this study, most of the subjects had experienced setback in their study and athletic performance due to overtraining or burnout. The findings highlight the need to empower and educate not only the athlete but also the coach on the impact of overtraining. Key findings from this study were the need for open coach–athlete communication, in-depth planning and the need for adequate recovery.
Another important paper, where the authors recommend and highlight the need of development of a special training program for student-athletes aimed at development and improvement of both sport skills and academic results, is:
Uzunov, M., Otcheva, G. Relationship of the psychological factors in the performance of technical elements by students from Technical University of Sofia practicing table tennis in classes of physical education. SocioBrains, No. 51, 2018, 366-375.
I recommend to the authors to extend the introduction by including the above and other related sources.
Section 2 is dedicated to the Materials and Methods used in the research. The sample studied is described very well. But again, the question of gender skewness of the results must be addressed, as mentioned above and addressed by the references which I mentioned.
The protocol for monitoring the training loads, the wellness questionnaire, sRPE and CMJ are excellently presented and summarized in Table 1.
The results are presented in Section 3. The interview questions are suitable and well-chosen. The descriptive analysis of the perception of completing the sRPE and the descriptive analysis of perception of completing the CMJ task are presented in Table 4 and Table 5. While the descriptive analysis of the effects of academic demands on performance and sport performance on academic demands is presented in Table 6. The results are correct and supported by other studies.
The authors have made an excellent discussion of the results. The conclusions drawn by the authors are supported by the results and confirmed by other researches, as indicated.
I recommend to the authors, in their future work to study the problem of uncertainty in the students opinion. This problem can be modelled through the use of fuzzy sets, or their extensions – the intuitionistic fuzzy sets. These tools allow to quantify the uncertainty in the students opinion and would give better results.
Overall, the paper is of high quality and presents some significant results. I recommend that the paper be published once the authors address adequately my critical remarks.
There are a few unclear sentences. For example, the sentences between lines 54-56; 98-99; 109-111, etc., are unclear. There are also a few grammar errors. Some sentences are too long and should be split.
Reviewer 2 Report
Introduction
- The authors have provided a comprehensive theoretical framework on FOMO and FOBO in market selling in their work that deserves significant praise. Their thorough exploration of the relevant literature, combined with their original insights, has resulted in a highly informative and valuable contribution to the field. The dedication is evident in the quality of their work.
- To improve the accuracy and reliability of their research, it is recommended that the authors provide citations for some bold statements throughout the manuscript (e.g., lines 54-56).
- While it is true that exploring new research questions is an important aspect of scientific inquiry, a lack of prior research in a particular area is not necessarily a compelling reason for conducting a study. Instead, researchers should identify a specific gap in the literature that their study aims to address and explain why this gap is important. Moreover, simply stating that a relationship between variables has not been explored before (line 84) does not provide a clear rationale for the study or indicate its potential contributions to the field.
- Overall, the introduction section of your manuscript appears to contain general statements that do not create great interest in your research topic since variables under analysis, mainly emotion of nurse students, are overwhelmingly cited and analyzed. Compelling arguments should be presented to better engage readers and demonstrate the importance of your study. As it currently stands, your research is another study with significant limitations that limits interpretation and practicality.
Methods
- The authors should disclose sampling method for transparency.
Results
- Nothing to report.
Discussion
- The authors have done an excellent job of presenting their findings in a clear and accessible manner. Their discussion of the results is thorough and insightful, providing a comprehensive analysis of the data. Additionally, the practical implications of their study are well-developed and provide valuable insights for practitioners and policymakers in the field.
- However, the authors' research should remark the significant contribution to the literature namely implications for practice.
Conclusion
- Should disclose more clearly the quantitative and qualitative (mix-method research) results of the study in a narrative fashion.
needs revisions.
Reviewer 3 Report
I would like to thank the Authors for the opportunity to review the article submitted to the journal. I have no slightest doubt that the article is of great quality and high value. The metodical, the results section as well as the discussion are very well expressed.
I only have a one suggestion, i.e. I recommend to use "sex" since "gender" is a much more ambiguous concept today. See, for example:
a) https://medicine.yale.edu/news-article/what-do-we-mean-by-sex-and-gender/
b) Rioux, C., Paré, A., London-Nadeau, K., Juster, R. P., Weedon, S., Levasseur-Puhach, S., ... & Tomfohr-Madsen, L. M. (2022). Sex and gender terminology: a glossary for gender-inclusive epidemiology. J Epidemiol Community Health, 76(8), 764-768.
c) Torgrimson, B. N., & Minson, C. T. (2005). Sex and gender: what is the difference?. Journal of Applied Physiology, 99(3), 785-787.
d) Muehlenhard, C. L., & Peterson, Z. D. (2011). Distinguishing between sex and gender: History, current conceptualizations, and implications. Sex roles, 64(11), 791-803.
Congratulations of your hard work.
Best regards
In my opinion only minor editing of English language is reguired.
Round 2
Reviewer 1 Report
I would like to thank the authors for taking into account my recommendations. The paper has been significantly improved. I recommend that the paper be published in the present form.
Some sentences should be editted to become clearer.
Reviewer 2 Report
The authors did an excellent job reviewing their manuscript.
Minor spelling check.